# Exosome Cargos as Biomarkers for Diagnosis and Prognosis of Hepatocellular Carcinoma

**DOI:** 10.3390/pharmaceutics15092365

**Published:** 2023-09-21

**Authors:** Yulai Zeng, Shuyu Hu, Yi Luo, Kang He

**Affiliations:** 1Department of Liver Surgery, School of Medicine, Renji Hospital, Shanghai Jiao Tong University, Shanghai 200127, China; oliverchina@outlook.com (Y.Z.); hushuyu0904@sjtu.edu.cn (S.H.); 2Shanghai Engineering Research Center of Transplantation and Immunology, Shanghai 200127, China; 3Shanghai Institute of Transplantation, Shanghai 200127, China

**Keywords:** exosome, hepatocellular carcinoma, biomarker, microRNA, long noncoding RNA, circular RNA

## Abstract

Hepatocellular carcinoma (HCC) is one of the most common cancers worldwide. Due to the insidiousness of HCC onset and the lack of specific early-stage markers, the early diagnosis and treatment of HCC are still unsatisfactory, leading to a poor prognosis. Exosomes are a type of extracellular vesicle containing various components, which play an essential part in the development, progression, and metastasis of HCC. A large number of studies have demonstrated that exosomes could serve as novel biomarkers for the diagnosis of HCC. These diagnostic components mainly include proteins, microRNAs, long noncoding RNAs, and circular RNAs. The exosome biomarkers showed high sensitivity and high specificity in distinguishing HCC from health controls and other liver diseases, such as chronic HBV and liver cirrhosis. The expression of these biomarkers also exhibits correlations with various clinical factors such as tumor size, TMN stage, overall survival, and recurrence rate. In this review, we summarize the function of exosomes in the development of HCC and highlight their application as HCC biomarkers for diagnosis and prognosis prediction.

## 1. Introduction

Hepatocellular carcinoma (HCC) ranks as the fifth most prevalent form of cancer globally and stands as the third leading cause of cancer-related mortality on a global scale, accounting for at least 780,000 deaths each year [1,2]. HCC commonly manifests in individuals with chronic liver disease, such as those afflicted with hepatitis B or C viral infections (HBV or HCV), alcohol misuse, and nonalcoholic fatty liver disease [3]. For patients with HCC in an early stage, surgical resection is the standard of care, and the prognosis is relatively promising [2]. Nevertheless, as a result of the covert nature of HCC initiation and the absence of distinct biomarkers in the early stages, the majority of patients receive a diagnosis when the disease has already progressed to its advanced stages, leading to an unfavorable prognosis [4]. For example, the classic blood diagnostic marker alpha-fetoprotein (AFP) is widely used, but its sensitivity and specificity are not satisfactory, with a sensitivity of 41–64% and a specificity of 80–94% [5]. The primary treatment strategies for HCC are surgical excision, liver transplantation, local delivery of radiation or chemical medicines, and combination therapy [2]. However, they are restricted by various factors such as the stage of cancer at the time of diagnosis, the presence of residual tumors, and the development of drug resistance [2]. Consequently, the overall efficacy of these treatments is still not ideal, with the five-year survival rate remaining at 18% [6,7]. Considering the circumstances, the identification of noninvasive, effective, and specific diagnosis biomarkers for early diagnosis and prognosis prediction is vital for the proper management of HCC.

Among noninvasive diagnosis methods, the utilization of liquid biopsy, a technique that involves the collection and analysis of samples derived from bodily fluids like blood and urine, has garnered significant interest and opened new avenues for the detection of cancer [8,9]. Circulating tumor cells, circulating tumor DNA, and exosomes are now recognized as the primary aspects of a liquid biopsy [8]. Exosomes, which represent a novel frontier in this field, offer distinct advantages over other liquid biopsies in terms of accessibility and the comprehensive information they provide about living cells and tumor microenvironments [9]. The exosome is a type of extracellular vesicle (EV) secreted from intracellular multivesicular bodies (MVBs) into the extracellular space. Through the transmission of the containing components, exosomes take part in the intercellular communication and remodeling of the extracellular matrix [10]. Exosomes have been proven to play a vital role in the onset, development, diagnosis, and treatment of HCC in recent vivo and in vitro experiments [11]. Related research has provided evidence of alterations in the composition of exosomes derived from HCC cells. Moreover, these changes have been found to contribute to the progression, metastasis, and immunosuppression of the cancer [11,12,13]. The assessment of the abundance of specific exosome components has led to the increased recognition of exosomes as promising biomarkers for the early diagnosis and prognostication of HCC.

## 2. Exosomes and HCC

The majority of human cells have the capability to generate a specific lipid membrane-enclosed vesicle known as an extracellular vesicle (EV). These vesicles have the ability to be released from the cells and exist outside of them. EVs can be classified into three primary types based on their biogenesis and diameter: microvesicles/ectosomes, exosomes, and apoptosomes/apoptotic bodies [14]. According to Minimal information for studies of extracellular vesicles 2018 (MISEV2018) [15], authors are strongly encouraged to employ operational words to describe different subtypes of extracellular vesicles (EVs) instead of using names like exosome and microvesicle, which have a long history of multiple and conflicting meanings, as well as incorrect assumptions about their distinct biogenesis processes. However, due to the lack of consensus in previous studies in this field, we still use exosomes in this paper to refer to small membranous vesicles ranging in size from 30 to 150 nm, with a density between 1.10 and 1.21 g/mL [10,14]. Ultracentrifugation is widely regarded as the gold standard technique for exosome separation, and it is commonly employed in research studies [9].

Exosomes are produced by almost all cells of diverse organisms and distributed widely in bodily fluids such as blood [16], saliva [17], milk [18], and urine [19]. Covered in a phospholipid bilayer membrane, exosomes contain a variety of cellular components such as proteins, microRNAs (miRNAs/miRs), long noncoding RNAs (lncRNAs), circular RNAs (circRNAs), messenger RNAs (mRNAs), and DNAs. Exosomes are recognized as a significant form of intercellular communication, facilitating the transfer of substances between cells and tissues. This process involves the transport of vesicle contents, thereby conveying information and serving as a vital signaling mechanism [10]. They play a crucial role in the physiological processes of cellular formation, growth, differentiation, and aging in human cells, particularly in relation to the development, metastasis, and progression of cancer [20]. Numerous studies have substantiated the exact regulation of exosome synthesis, secretion, transport, uptake, and release through unique signaling pathways, which in turn exert an influence on the growth and immunosuppression of HCC [21]. In this section, we explore the mechanisms by which HCC cells generate and release exosomes to facilitate intercellular communication and modulate microenvironmental conditions, as well as the composition and functions of these extracellular vesicles (Figure 1). In order to elucidate the impact of exosomes on HCC, we carefully chose a subset of exosome molecules that are both representative and influential, as well as incorporating the most recent research findings to explain the status of current research.

### 2.1. Formation and Secretion of Exosomes

Exosomes originate from early endosomes, which are formed by the engulfment of the cell membrane and encapsulation of intracellular materials such as target proteins, nucleic acids, and lipids. With the collection, modification, and sorting of these materials, early endosomes mature into MVBs (also known as late endosomes) containing intraluminal vesicles (ILVs) [13]. MVBs can undergo constitutive fusion with the cell membrane, a process mediated by the trans-Golgi network. Likewise, it can also be analytically confused with the cell membrane via mechanisms such as receptor activation or alterations in ion concentration. These processes can result in the secretion of exosomes or the degradation of MVBs by lysosomes, depending on their respective functions and direction of movement [22]. The principal processes accountable for the generation of exosomes encompass the endosomal sorting complex required for the transport (ESCRT) pathway and the ESCRT-independent pathway [23]. The ESCRT pathway primarily encompasses the involvement of five distinct types of proteins, namely ESCRT-0, ESCRT-I, ESCRT-II, ESCRT-III, and vacuolar protein sorting related protein 4 [23]. In addition, the process also involves the participation of small molecule 4-transmembrane proteins, cytoskeleton proteins (such as actin and microtubules), molecular motors (including dynein, kinesin, and myosin), and molecular switches (such as small GTPase) [10,24].

### 2.2. Exosomes Participate in the Biological Regulation of HCC Cells

Research indicates that HCC cells have been observed to release exosomes at an elevated level [25]. Furthermore, the composition of these exosomes varies under different pathological and physiological circumstances [26]. Tumor-derived exosomes have the capability to transport their cargo to neighboring and remote cells, hence exerting an influence on the growth and proliferation of these recipient cells [13]. The PTEN gene is recognized as a prominent anti-oncogene, with its expression being observed to be diminished in several tumor types, including HCC [27]. Exosomal miR-21 generated from HCC cells modulates the expression of tumor suppressor genes PTEN and PTENp1 through several mechanisms, hence influencing the proliferation of HCC cells [28]. Gai et al. [29] revealed that exosomes enriched with Golgi membrane protein 1 (GOLM1) triggered the GSK-3β/MMPs signaling axis in neighboring cells, resulting in enhanced cell proliferation and migration. Moirangthem A et al. [30] provided evidence suggesting that miR-126-3p played a significant role in influencing the migration, invasion, and spheroid formation of tumor cells when hematopoietic stem cells are present. The exosomes of HCC cells could release miR-122, which is internalized by miR-122-deficient HCC cells, thereby inhibiting HCC growth through the suppression of cell cycle progression [31].

A cancer-associated fibroblast (CAF) is an integral constituent of the tumor extracellular matrix, playing a significant role in HCC metastasis. The markedly increased expression of miR-1247-3p in HCC exosomes can lead to the downregulation of β-1,4-galactosyltransferases III (B4GALT3), activating the integrin β1/NF-κB pathway [32]. These processes will lead to the transformation of fibroblasts into CAFs, which can cause the secretion of inflammatory factors such as IL-6 and IL-8 to the tumor extracellular matrix, promoting HCC progression [32].

### 2.3. Exosomes Promote the Metastasis and Progression of HCC

Multiple studies have demonstrated the significant involvement of exosomes in the metastasis and progression of HCC [11]. Exosomes induce physiological alterations in cells caused by facilitating the transportation of molecules, hence influencing cellular differentiation and migration. These exosomes exert their influence through various mechanisms, such as directly stimulating tumor growth, conferring migratory and invasive capabilities to cells with limited or no metastatic potential, inducing epithelial–mesenchymal transition (EMT), restoring premetastatic microenvironment, and promoting angiogenesis to establish a favorable environment for metastatic potential [1,24]. Previous studies have demonstrated that MET proto-oncogene, S100 family members, and caveolins mobilize normal hepatocytes and participate in the migration and progression of HCC [33]. Additionally, it has been demonstrated that Interleukin-6 (IL-6) and Golgim1 play a crucial role in promoting the invasion of HCC by releasing exosomes [34,35]. Several studies have reported that S100A4 [36] and CircPTGR1 [37] are identified as the main components of exosomes derived from highly metastatic HCC cells. These components have been observed to exert an influence on neighboring cells, hence promoting the metastatic potential of low-metastatic HCC cells.

EMT is a biological process of cellular transformation characterized by the loss of epithelial traits and acquisition of mesenchymal features, endowing cells with the ability to metastasize and invade, thereby contributing to tumor progression and metastasis [38]. Exosomal circRNA hsa_circ_0003288 could act on miR-145, which regulates PD-L1 expression, promoting EMT, migration, and the invasiveness of HCC [39]. The overexpression of the host gene matrix metallopeptidase 2 was observed as a result of exosome-derived circ_MMP2 (also known as has_circ_0039411), which acts as a sponge for miR-136-5p. This interaction has been found to cause the spread of HCC and is associated with a worse overall survival rate in HCC patients [40]. The upregulation of miR-4669 resulted in the increased migratory capacity of HCC cells and conferred resistance to sorafenib treatment, accompanied by an upregulation of sirtuin 1 [41]. Zhang et al. [42] observed that exosomal circ_0046600 could facilitate the advancement of HCC by sequestering miR-1258 and subsequently upregulating SERBP1.

In recent years, many studies have found that HCC cells promote extracellular environment inflammation and angiogenesis through exosomes, in order to provide a suitable environment for metastasis. The exosome contents involved in the process include RNAs, such as miR-210 [43], miR-155 [44], lncRNA-H19 [45], and proteins, such as CLEC3B [46], ANGPT2 [47], HSP70 [48], etc. The study conducted by Lin et al. [43] revealed that HCC cells release exosomes containing miR-210, which may be taken up by endothelial cells. This process facilitates tumor angiogenesis by specifically targeting the signal transducer and activator of transcription 4 and the signal transducer and activator of transcription 6. Dai et al. [46] demonstrated that exosomes exhibiting reduced levels of CLEC3B were found to enhance the migratory, invasive, and epithelial–mesenchymal transition capabilities of HCC tumor cells and endothelial cells.

### 2.4. Exosomes Involve in the Immunosuppression of HCC

The formation of HCC undergoes accumulating gene mutations and the interaction of the tissue environment. In normal circumstances, the immune system may recognize the abnormalities in the early stage and stop this process. However, in cancer tissue, tumor cells possess the ability to deceive immune cells and escape immune surveillance, resulting in immunosuppression and cancer development [49]. The effects of exosomes on HCC encompass decreasing immune cell activity, facilitating the apoptosis of CD8+ T lymphocytes, eliciting the activation of regulatory T cells (Tregs), and leading to Th17/Treg imbalance [1,12,21,24]. Several studies found that the highly abundant exosomal 14-3-3 ζ protein can inhibit the activation, proliferation, and differentiation of T cells, inducing their transformation into Tregs, which causes protumor immune response [50]. The interaction between PD-L1 expressed in tumor cells and PD-1 present on the surface of CD8+ T cells leads to the induction of apoptosis, exhaustion, and the inactivation of CD8+ T cells [51]. PD-L1 is also observed to be present on the external membrane of exosomes produced from HCC cells [52]. Hu et al. [53] found that the release of exosomal circCCAR1 by HCC cells played a role in immunosuppression. Specifically, it facilitates the malfunction in CD8+ T cells within the context of HCC and leads to resistance against anti-PD1 immunotherapy. Huang et al. [54] revealed that exosomal circGSE1 plays a significant role in advancing the progression of HCC. This effect is achieved via the induction of regulatory T cells (Tregs) growth, which is mediated by the regulation of the miR-324-5p/TGFBR1/Smad3 axis.

In addition to specific immune cells, tumor-cell-derived exosomes also intensively affect other immune cells. The involvement of exosomes in the suppressive innate immune milieu of HCC includes their impact on natural killer cells, dendritic cells, and macrophages. Tumor-associated macrophages can differentiate toward the M2 phenotype, inducing tumor immunosuppression and promoting the metastasis and progression of the tumor. Studies have shown that HCC-derived exosomes are abundant in lncRNA-TUC339 [55], circTMEM181 [56], miR-23a-3p [57], and miR-146a-5p [58]. These constituents facilitate the polarization of M2 macrophages, thereby hindering the efficiency of CD8+ T cells. Ye et al. found the growth of regulatory B cells expressing TIM-1 is promoted by the tumor-derived exosomal HMGB1, hence facilitating immune evasion in HCC [59]. Additionally, it was observed that the level of exosomal circUHRF1 increased in HCC cell tissues. This elevation was found to impede the production of interferon-gamma and tumor necrosis factor-alpha via natural killer cells, thus resulting in the impaired functionality of NK cells [60].

## 3. Exosome Cargos as Biomarkers for HCC

HCC is a tumor with a high degree of malignancy, rapid progression, and poor prognosis [1]; therefore, it needs reliable biomarkers for early diagnosis, prognosis, and the evaluation of treatment efficacy. However, ultrasonography and existing biomarkers such as des-gamma-carboxy prothrombin (DCP) and glypican-3 are not so satisfactory [61]. Additionally, histological testing is restricted by their invasiveness, and imaging methods such as computed tomography (CT) and magnetic resonance imaging (MRI) are limited in their detection of small tumors [62]. Many studies have suggested that exosomes are novel, noninvasive biomarkers for cancer detection. Compared to other indicators, exosomes are stable in blood and other body fluids, having the advantages of minimal invasiveness and easy sample acquisition [9]. The expression of exosome cargos has the potential to differentiate HCC patients from those with other liver diseases and normal controls. Additionally, exosome cargos may exhibit a significant association with several clinical parameters such as tumor size, TNM stage, portal vein tumor thrombosis, overall survival, and other relevant factors, presenting a better characteristic than AFP. We summarize some representative exosome components that could serve as valuable diagnosis and prognosis predictors of HCC in future studies (Table 1).

### 3.1. MicroRNAs

Exosomal microRNAs (miRNAs/miRs) exhibit greater stability compared to serum-free miRNAs, owing to the lipid bilayer’s protection of exosomes [63]. Furthermore, these miRNAs are specifically abundant within exosomes, rendering them significant in the control of genes, translation processes, and epigenetic processes associated with HCC [64]. MiRNAs make complex interactions between mRNA, lncRNA, and circRNA, and exosomes enhance miRNAs’ ability to crosstalk between microenvironment and HCC cells. These interactions make them critical to metastasis, proliferation, and angiogenesis of HCC, which heavily rely on the communication between different cells. Because of these unique properties, exosomal miRNAs are ideal diagnostic markers and prognostic factors for HCC patients, thereby having become one of the most studied exosomal cargos.

MiR-21 is a prominently expressed miRNA in HCC, contributing to the proliferation, metastasis, and chemotherapy drug resistance of HCC cells [24]. Wang et al. [65] discovered that the expression of exosomal miR-21 exhibited a notable increase in the group with HCC in comparison to the chronic HBV group. This increase in expression was also found to be positively associated with the presence of cirrhosis and a more advanced stage of tumor development. More importantly, the sensitivity of exosomal miR-21 is much higher than serum miR-21 in the diagnosis of HCC. This finding revealed the potential of exosomal miRNAs as valuable biomarkers, inspiring further research in this field. Similarly, exosomal miR-10b-5p mediates cell communication and facilitates cell proliferation [66], and it has been recently identified as a novel biomarker of early-stage HCC diagnosis. Exosomal miR-10b-5p was overexpressed in early-stage HCC patients from a validation cohort (consisting of 60 chronic liver disease patients, 90 HCC patients, and 28 healthy controls) with 0.934 area under the curve (AUC) [67]. Wang et al. [68] screened exosomal miRNAs that exhibit differential expression patterns between HCC and liver cirrhosis. After this process, exosomal miR-122 and miR-148a underwent further analysis. The study discovered that exosomal miR-48a performed significantly better than AFP in distinguishing HCC from liver cirrhosis (with an AUC of 0.891 compared to 0.712) but failed to discriminate HCC from chronic hepatitis. Furthermore, the combination of miR-122, miR-148a, and AFP gained the highest AUC (0.990) for differentiating HCC from normal controls [68]. In the study conducted by Ghosh et al. [69], the AFP levels in 94% of HCV-HCC and 62% of HBV-HCC patients did not reach 250 ng/mL, exhibiting poor diagnostic utility. However, the combination of four exosomal miRNAs—miR-10b-5p, miR-21-5p, miR-221-3p, and miR-223-3p—was significantly effective in differentiating low-AFP HCC from other liver diseases, with an AUC of 0.80. Other biomarkers such as exosomal miR-34a [70], exosomal miR-483-5p [71], and exosomal miR-10b-5p [67] have also been proven to have better sensitivity and specificity than serum AFP.

Exosomal miRNAs could also be used as a prognosis factors for HCC, predicting treatment efficacy and overall survival. Wei et al. [64] extracted serum exosomes from 90 HCC patients and analyzed exosomal miR-370-3p and miR-196a-5p expression with disease characteristics. The results show that these two exosomal miRNA levels were positively correlated with tumor grade, TNM stage, and the prognosis of HCC patients, exhibiting similar properties, such as alanine aminotransferase and aspartate aminotransferase. The overexpression of miR-92b could promote the migration of HCC and downregulate NK-cell-mediated cytotoxicity [72]. Nakano et al. [73] confirmed a higher level of exosomal miR-92b in HCC patients before living donor liver transplantation. The continued increase in exosomal miR-92b expression after transplantation was associated with a higher post-transplant HCC recurrence rate. The same occurred for exosomal miR-718, whose decreased expression could also indicate tumor recurrence after liver transplantation [25]. Exosomal miR-122, which is described above, can also serve as a predictor for HCC treatment efficacy, especially for transarterial chemoembolization (TACE). Suehiro et al. [74] demonstrated that the level of exosomal miR-122 was significantly decreased after TACE, while another diagnostic biomarker, exosomal miR-21, underwent no change. The ratio of serum exosomal miR-122 after TACE/before TACE was positively associated with longer disease-specific survival, which presented a potential clinical application.

### 3.2. Long Noncoding RNAs

Long noncoding RNAs (lncRNAs) are RNA molecules with a transcript length of more than 200 nt that lack the ability to encode proteins. They interact with proteins, RNA, and DNA in their unique subcellular localization to regulate gene expression [75]. LncRNAs could be encapsulated within exosomes, thereby preserving them against degradation and facilitating their transmission to neighboring or remote cells [76]. As important biological regulators in exosomes, exosomal lncRNAs are highly capable of influencing multiple molecular signaling pathways and facilitating controlling transcription across different cells via chromatin modification. As a result, they can promote the proliferation, migration, and metastasis of HCC, offering a novel avenue for the diagnosis and treatment of hepatocellular carcinoma [77,78].

Huang et al. [79] demonstrated that 8572 lncRNAs were differentially expressed in plasma exosomes of HCC patients. Among these lncRNAs, exosomal RP11-85G21.1 regulated the expression of miR-324-5p, promoting HCC cellular proliferation and migration. More importantly, the expression level of exosomal RP11-85G21.1 can not only distinguish AFP-positive HCC patients from non-HCC patients, but it is also serves as an effective AFP-negative HCC biomarker, with AUC values of 0.883 and 0.869, respectively. LncRNA THEMIS2-211 could interact with miR-940 and upregulated SPOCK1 expressions, therefore behaving as an oncogene that promotes the metastasis and migration of HCC [80]. Yao et al. [80] found that the performance of exosomal THEMIS2-211 is superior to AFP in the diagnosis of stage I HCC patients (with an AUC of 0.818 compared to 0.731) and its upregulated level was associated with the poor prognosis of HCC. These studies indicate that exosomes lncRNAs have better properties than AFP in the diagnosis of HCC. Sun et al. [81] analyzed expression levels of eight exosomal lncRNAs in HCC patients and healthy controls by using qRT-PCR, and their specificity and stability were evaluated. Their research proved that the upregulated exosomal LINC00161 has excellent stability and specificity in distinguishing HCC patients from healthy controls. Furthermore, LINC00161 expression was positively correlated with serum AFP concentration and the TNM stage [81], indicating that exosomal lncRNAs could reflect additional clinical characteristics. Liao et al. [82] analyzed the relationship between exosomal MALAT1 and the prognosis of HCC. The results show that the high expression of exosomal MALAT1 was significantly related to shorter progression-free survival and shorter overall survival, which exhibited moderate predictive power (AUC range from 0.661 to 0.731). These studies revealed the potential prognosis usage of lncRNAs.

In addition, certain researchers have developed diagnostic panels that incorporate various exosomal lncRNAs. For example, Kim et al. [83] found that exosomal MALAT1, SNHG1, DLEU2, and HOTTIP could serve as promising biomarkers in the diagnosis of HCC. Moreover, the panel combining exosomal DLEU2 and AFP achieved the best positivity (96%) in the diagnosis of early HCC, and the combination of exosomal MALAT1 and exosomal SNHG1 exhibited the highest AUC (0.899). Xu et al. [84] recruited 301 participants, including the HCC group, liver cirrhosis group, chronic hepatitis B group, and healthy controls. Researchers isolated participants’ serum exosomes and measured their levels with TaqMan PCR. The results of this study revealed that the levels of exosomal ENSG00000258332.1 and LINC00635 were significantly higher in the HCC group than in others, and they achieved the highest diagnostic value when these two exosomal lncRNA levels were combined with serum AFP.

### 3.3. Circular RNAs

Circular RNAs (circRNAs) are a new type of non-coding RNAs whose structures are covalently closed loops formed from single-stranded RNAs [61]. CircRNA has recently been identified as a key regulatory factor in HCC and attracted increasing attention. Due to its unique closed-loop structure, circRNA cannot be easily degraded by exonuclease [85]. Furthermore, when circRNA is enclosed within exosomes, its stability is further enhanced [86]. With its stability and function of regulating miRNA splicing, exosomal circRNAs play a crucial role in the progression, immune escape, invasion, and drug resistance of HCC [87,88,89]. Recent studies have also shown that exosome circRNAs possess the ability to be novel biomarkers for the diagnosis and forecasting of HCC.

Lyu et al. [90] found that exosomal circ_0070396 was upregulated in HCC patients, and it presents a higher diagnostic value than AFP (with an AUC of 0.86 compared to 0.72 for distinguishing HCC from healthy donors). A combination of AFP and circ_0070396 performs better than a single marker in the differentiation of HCC from healthy donors, chronic HBV, and liver cirrhosis patients, with AUC values of 0.94, 0.85, and 0.75, respectively. Lin et al. [91] reported that exosomal circ-0072088 suppresses the invasion and migration of HCC cells by regulating MMP-16, and it is upregulated in HCC patient plasma exosomes, which makes it a novel biomarker for diagnosis. Moreover, a high level of exosomal circ-0072088 indicates a poor prognosis for HCC patients. Wang et al. [92] reported that circ_0028861 is downregulated in HCC patients compared to chronic HBV and liver cirrhosis. It is more significant than AFP in the diagnosis of small, early-stage, and AFP-negative HCC, with AUC values of 0.81, 0.82, and 0.78, respectively. Chen et al. [93] detected the circ-0051443 expression level in 60 HCC patients and 60 healthy controls and discovered that the expression in HCC was significantly lower than in controls, so circ-0051443 could be a valuable biomarker in the diagnosis of HCC.

Exosomal circAKT3 is a prognosis indicator for HCC patients. Luo et al. [94] showed that exosomal circAKT3 levels not only significantly increased in HCC compared with healthy controls but also positively correlated with higher tumor recurrence rates and higher mortality. Lin et al. [95] compared the expression of exosomal circ-G004213 during transarterial chemoembolization and confirmed that exosomal circ-G004213 was positively associated with the prognosis of HCC patients after transarterial chemoembolization. Moreover, its level can serve as a predictor of HCC patients’ cisplatin resistance. Another study conducted by Zhang et al. [60] demonstrated that exosomal circUHRF1 induces natural killer cell exhaustion. The patients with upregulated exosomal circUHRF1 showed resistance to anti-PD1 immunotherapy and a high cumulative recurrence.

Some novel exosomal circRNAs could also be recognized as potential biomarkers whose diagnostic values have not been clearly examined. The upregulated exosomal circTGFBR2 promotes ATG5-mediated protective autophagy to enhance the resistance to starvation stress of HCC cells [96], and researchers indicated that circTGFBR2 could be a potential biomarker and therapeutic target for HCC. Exosomal circCCAR1 is another circRNA that displays a significant upregulation in HCC tumor tissues [53]. This upregulation has been found to play a role in the impairment of CD8+ T-cell activity inside HCC, thereby leading to resistance against anti-PD1 immunotherapy [53]. The findings of this study suggest that exosomal circCCAR1 has potential as a diagnostic marker and predictor of immunotherapy efficacy. However, further investigation is required to validate this conclusion.

### 3.4. Proteins

Exosomal proteins have the potential to serve as suitable biomarkers for HCC detection. They are abundant in HCC-cell-derived exosomes. He et al. detected 213 exosomal proteins derived from HCC cell lines using mass spectrometry [33], and highly aggressive HCC tends to release critical proteins via exosomes [97]. Due to the abundance of exosomal proteins and the unique role that they play in protein communication, exosomal signal pathways are important for the metastasis, progression, and immunosuppression of HCC. Studies have shown that exosomal proteins may become potential therapeutic targets [77] and ideal diagnostic markers of HCC.

Hepcidin is an essential iron regulator in hepatocytes, which blocks iron exports by binding to ferroprotein to maintain iron concentration in extracellular fluid [98]. Sasaki et al. [99] utilized a modified method using a two-step PCR amplification system to detect exosomal hepcidin mRNA levels in HCC patients and patients of other chronic liver diseases. Researchers found that its levels in HCC were significantly higher than in chronic liver diseases and healthy people, indicating that exosomal hepcidin mRNA could be a diagnostic marker to distinguish them. Arbelaiz et al. [100] found that exosomal galectin-3 binding protein (G3BP) was notably elevated in HCC patients compared to healthy controls and cholangiocarcinoma patients, with an AUC of 0.904 and 0.894, respectively. Several exosomal proteins have the potential to serve as prognostic indicators, allowing for the prediction of survival and recurrence rates among patients with HCC. S100A4 is an essential component in HCC exosomes that enhances tumor metastasis through the activation of STAT3 and the induction of osteopontin production [36]. The level of exosomal S100A4 was examined by researchers in relation to survival and recurrence rates. It was observed that the combination of exosomal S100A4 levels and osteopontin levels exhibited a more favorable predictive performance compared to AFP [36]. Fu et al. [101] reported that exosomal SMAD3 was abundant in the serum of patients with HCC, inducing tumor cells’ adhesive ability. In addition, exosomal SMAD3 levels were positively correlated with disease stage and pathological grade; therefore, scholars have indicate that exosomal SMAD3 may have diagnostic and prognostic values. Although the function of HCC exosomal proteins has been widely researched, their diagnostic values have not been fully noticed and uncovered, which provides possible research directions. Wang et al. [102] isolated exosomes from different HCC cells to characterize the function of exosomal proteins and perform protein profiling. Among all the proteins, adenylyl cyclase-associated protein 1 (CAP1) was correlated with HCC metastasis and significantly evaluated in exosomes. Thus, exosomal CAP1 was suggested to be a potential diagnostic factor of HCC and worthy of further study.

### 3.5. Others

Aside from miRNAs, lncRNAs, circRNAs, and proteins, other exosomal cargos such as DNA and mRNA, also play critical roles in tumor progression, which can serve as indicators for the diagnosis and prognosis of HCC.

Exosomal DNA. Detecting mutant DNA using PCR-based sequencing techniques is a substantial part of liquid biopsies. Circulating cell-free DNA (cfDNA) is a potential biomarker for tumor diagnosis, but the upregulated level is only detectable when hepatocytes have incurred damage. This poses a challenge in the early identification of HCC [103,104]. Exosomal DNA is protected by a lipid bilayer structure and exhibits a notable enrichment within these extracellular vesicles [105]. Compared to cfDNA, exosomal DNA has a higher molecular weight and better stability [106], and its mutations are more detectable than cfDNA [107]. Li et al. [105] detected a 747 G > T mutation in the TP53 gene inside exosomal DNA derived from HCC patients. Compared to patients with a low-frequency 747 G > T mutation, the high-frequency mutation group had shorter median recurrence-free survival and poorer prognosis though they may present better pathological characteristics such as low APF levels [105]. This study provided evidence that exosomal DNA could function as a novel diagnosis and prognosis tool in HCC.

Exosomal messenger RNA (mRNA). Researchers have also discovered the diagnostic potential of exosomal mRNA in HCC. Xu et al. [108] demonstrated that exosomal hnRNPH1 mRNA was significantly increased in HCC patients compared to liver cirrhosis, chronic hepatitis, and a healthy control group. Moreover, the upregulated levels were correlated with other clinical factors, including the Child–Pugh score, portal vein tumor thrombosis, overall survival, and TNM stage.

## 4. Conclusions

Exosomes have been identified as significant contributors to the pathogenesis of HCC, exerting their influence through many mechanisms. These mechanisms include modulating cellular proliferation and differentiation, facilitating the metastatic potential and progression of HCC, as well as partaking in immunosuppressive processes. Moreover, exosomes could behave as highly effective biomarkers for the diagnosis and prognosis of HCC. Compared to histological testing, exosomal detection is noninvasive, dynamically monitored, and highly accepted by patients. Furthermore, exosomes are stable, abundant in blood, and encapsulate a multitude of disease-related attributes, outperforming conventional lipid biopsies such as circulating tumor cells. The exosome biomarkers showed a high sensitivity and high specificity in distinguishing HCC from health controls and other liver diseases, such as chronic HBV and liver cirrhosis. They also perform significantly well in the diagnosis of AFP-negative HCC, demonstrating a higher AUC than AFP in various circumstances. Furthermore, the levels of these biomarkers may exhibit correlations with various clinical factors such as tumor size, TMN stage, overall survival, and recurrence rate, which could serve as valuable markers for predicting treatment efficacy and prognosis. In clinical applications, it is also common to employ a combination of between 2 and 4 exosome biomarkers, typically in conjunction with AFP, exhibiting superior performance compared to individual indicators.

Nevertheless, there remain some issues that warrant consideration. First, exosome extraction takes a long time and requires multiple steps, and the isolation process is not standardized. More importantly, in most research concerning HCC and exosomes, there is a lack of methodologies to identify whether biomarkers are from exosomes or other extracellular vesicles. A lack of precise definitions and specific indicators of has resulted in confusion within this particular research domain. Second, although several exosome cargos show a potential diagnostic value, their molecular mechanisms are still unclear. This leads to controversial results in different circumstances (such as individual differences like age and gender) because of their diverse functions in the development of HCC. Third, most studies are still in the preclinical experimental stage. Potential exosome biomarkers are widely available but not detailed enough and thus require more extensive clinical samples.

Hence, further steps are required in future studies. There is a need to improve the isolation and separation technique in order to enhance exosomes’ clinical applicability and achieve greater accuracy. The combination of different isolation methods and the development of reagent kits could lead this field forward. To address controversial findings and individual differences, a comprehensive evaluation of the involvement of specific constituents of exosomes in the formation and development of HCC is warranted. To enhance exosomes’ clinical applications, exosomal biomarkers should be compared and combined with other diagnostic methods, not only AFP, but also traditional HCC biomarkers like DCP and glypican-3, or histopathological biopsies and imaging tests, which have rarely been examined. Despite its limitations, it is necessary for researchers to be persistent in expanding this field to determine the clinical applications of exosome biomarkers for HCC.

## Figures and Tables

**Figure 1 pharmaceutics-15-02365-f001:**
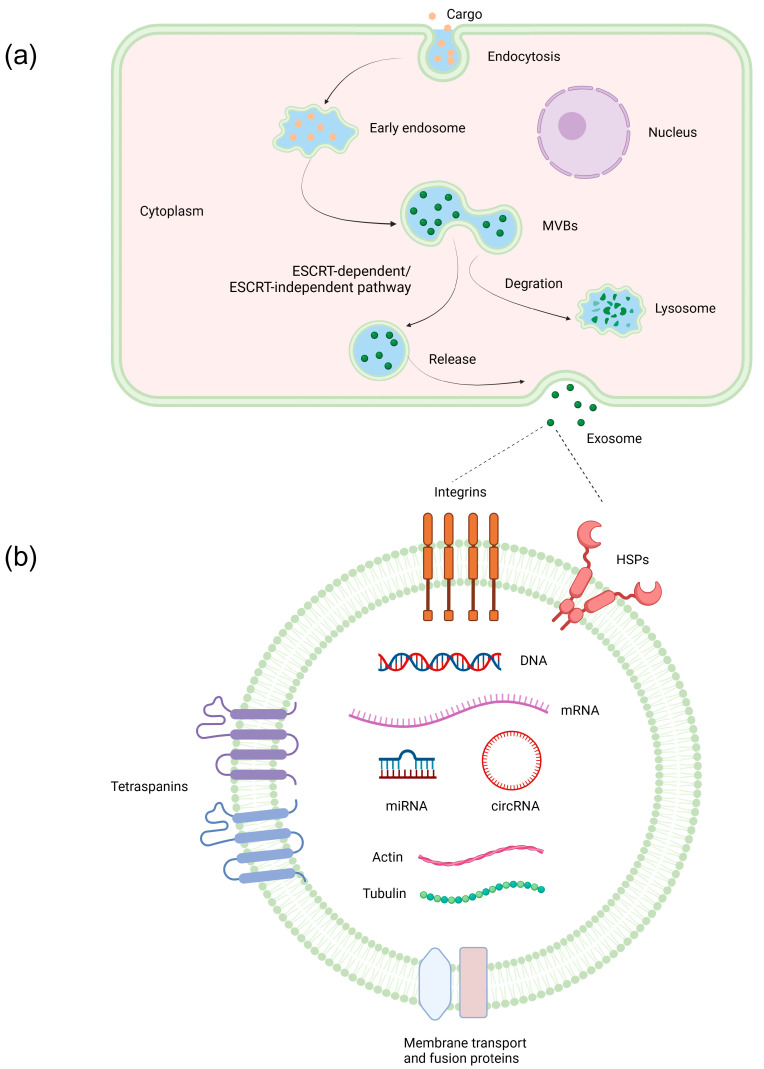
Schematic diagram of the formation and major components of exosomes. (**a**) The formation of exosomes. Endocytosis encapsulates materials to form early endosomes; early endosomes continue to mature into MVB; the intracellular transport and sorting of exosomes through the ESCRT-dependent pathway or ESCRT-independent pathway are carried out; the process of delivering MVB to lysosomes facilitates the degradation of exosomal cargo; exosomes anchor and subsequently fuse with the plasma membrane, resulting in the release of exosomes. (**b**) Major components of exosomes.

**Table 1 pharmaceutics-15-02365-t001:** The function and application of exosome biomarkers in HCC.

Components of Exosome	Biomarker	Function	Application	References
microRNA	miR-21	Contribute to the proliferation, metastasis, and chemotherapy drug resistance	Diagnosis	[53]
	miR-10b-5p	Mediates cell communication and facilitates cell proliferation	Diagnosis	[54]
	miR-122	Influence proliferation, migration, and invasion of HCC	Screening, diagnosis, and TACE efficacy prediction	[55,56]
	miR-92b	Promote the migration of HCC and downregulated the NK cell-mediated cytotoxicity	Prognosis: post-transplant HCC recurrence	[57]
	miR-718	Regulate the proliferation of HCC cells by mediating SEMA3B-AS1 and PTEN	Prognosis: post-transplant HCC recurrence	[58]
	miR-370-3p, miR-196a-5p	Drive tumor progression and immune evasion	Diagnosis and prognosis	[59]
	miR-34a	Promote proliferation, apoptosis, and autophagy of HCC cells	Diagnosis and prognosis: tumor infiltration depth and lymph node metastasis	[60]
	miR-483-5p	Promote HCC cells proliferation by downregulating CDK15	Diagnosis	[61]
Long noncoding RNA	LINC00161	Influence HCC progression	Diagnosis and prognosis	[62]
	SNHG1, DLEU2	Influence HCC progression	Diagnosis	[63]
	RP11-85G21.1	Promote HCC proliferation and migration by regulating miR-324-5p	Diagnosis: AFP+ and AFP- HCC	[64]
	ENSG00000258332.1	Influence HCC progression	Diagnosis and prognosis: lymph node metastasis and overall survival	[65]
	MALAT1	Regulate genes involved in DNA damage repair, homologous recombination, ferroptosis, infiltration of lymphocytes	Prognosis: progression-free survival and overall survival	[66]
	THEMIS2-211	Serve as an oncogene that promotes the proliferation, migration, invasion of HCC	Diagnosis and prognosis	[67]
Circular RNAs	circ_0070396	Influence HCC progression	Diagnosis	[68]
	circ-0072088	Suppresses invasion and migration	Diagnosis and prognosis: mortality	[69]
	circ_0028861	Influence HCC progression by regulating miRNAs and downstream tumor-related signaling pathways	Diagnosis: small, early-stage, and AFP-negative HCC	[70]
	circ-0051443	Promote HCC cell apoptosis and arrest the cell cycle	Diagnosis	[71]
	CircAKT3	Promote HCC progression by back splicing of AKT3 gene	Prognosis: recurrence rates and mortality.	[72]
	circ-G004213	Promotes cisplatin sensitivity	Prognosis: efficacy of transarterial chemoembolization	[73]
	circUHRF1	Induce natural killer cell exhaustion	Resistance to anti-PD1 immunotherapy and high cumulative recurrence	[52]
Protein	G3BP	Influence HCC progression by regulating mRNA translation and gene expression	Diagnosis	[74]
	S100A4	Promoted tumor metastasis	Diagnosis and prognosis: survival and recurrence rates	[28]
	SMAD3	Facilitate metastasis by regulating adhesion	Prognosis	[75]
	CAP1	Promoted tumor metastasis	Diagnosis	[76]
DNA	TP53 mutation	Promote tumorigenesis	Prognosis	[77]
Messenger RNA	hnRNPH1 mRNA	Associated with poorer differentiation of tumor cells	Diagnosis and prognosis: portal vein tumor thrombosis and overall survival	[78]

## Data Availability

Not applicable.

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
