# Peer review of "Exosome Cargos as Biomarkers for Diagnosis and Prognosis of Hepatocellular Carcinoma"

_pharmaceutics, 2023, doi:10.3390/pharmaceutics15092365_

Round 1
Reviewer 1 Report
The submitted manuscript is well-written and well-presented covering an interesting point on the function of exosomes in the development of Hepatocellular Carcinoma (HCC) and highlighting their application as HCC biomarkers for diagnosis and prognosis prediction.
However, there are some comments that I would like to share with the authors;
1- This interesting point has been well covered before in numerous review articles, for example;
- Nimitrungtawee N, Inmutto N, Chattipakorn SC, Chattipakorn N. Extracellular vesicles as a new hope for diagnosis and therapeutic intervention for hepatocellular carcinoma. Cancer Med. 2021 Dec;10(23):8253-8271.
- Yang S, Wang J, Wang S, Zhou A, Zhao G and Li P: Roles of small extracellular vesicles in the development, diagnosis and possible treatment strategies for hepatocellular carcinoma (Review). Int J Oncol 61: 91, 2022.
- Xue D, Han J, Liu Y, Tuo H, Peng Y. Current perspectives on exosomes in the diagnosis and treatment of hepatocellular carcinoma (review). Cancer Biol Ther. 2021 Apr 3;22(4):279-290.
I could not detect any difference in the coverage points from these previous review articles. What is the novelty presented in this submitted manuscript?
2- The article's coverage of studies investigating the biological function of exosomal content in HCC progression is limited. For example, exosomal proteins like Caveolin, MET, ITGαvβ5, Interleukins, and Golgim1 are well documented as efficient molecules for promoting the migration and invasion of HCC, but nothing is mentioned regarding their functional involvement in HCC progression.
The same for numerous verified exosomal non-coding RNAs. What criteria did the authors use to select the studies for their review? The inclusion and exclusion criteria for the studies chosen for coverage and citations must be specified in detail.
3- The sensitivity/specificity of exosomes as biomarkers in comparison with the current clinical biomarkers should be clearly presented and listed.
4- Limitations or pitfalls of using circulating exosomes as biomarkers should be clearly stated.
5- More details should be included for future research directions on the use of exosomes as diagnostic/prognostic biomarkers (I would suggest a separate paragraph with a headline).
6- A thorough scientific check should be run for the whole manuscript to avoid scientific inaccuracy. For example; on Page 4 line 110 it is stated that “PTEN is a significant oncogene” but in lines 111 and 112 it is stated that miR-21 targets the tumor suppressor genes PTEN !!
Reviewer 2 Report
The review of the manuscript “Exosomes as biomarkers for diagnosis and prognosis of hepatocellular carcinoma”
Yulai Zeng et al
In the review, the authors describe the role of exosomes as biomarkers for diagnosis and prognosis of hepatocellular carcinoma. The manuscript is written professionally in clear language. There are only some points to be corrected.
First. The statement “Exosomes as biomarkers” should be supported by use of exosomes as biomarkers. But it is not a case. Actually, the cargo of exosomes is used as a biomarker, not exosomes. Accordingly, the title should be changed.
Second.
There are some minor errors and typos. For instance:
P107. “Both the level and constitution of circulating exosomes in patients with HCC are notably changed in HCC patients”.
P110. “PTEN is a significant oncogene”. PTEN is a tumor suppressor (anti-oncogene).
But the major objection is that recently a lot of reviews covering exact the same topic was published.
Here, just some recent references, in addition to references mentioned in the manuscript.
1. Chen R, Xu X, Tao Y, Qian Z, Yu Y. Exosomes in hepatocellular carcinoma: a new horizon. Cell Commun Signal. 2019 Jan 7;17(1):1. doi: 10.1186/s12964-018-0315-1. PMID: 30616541; PMCID: PMC6323788.
2. Yang N, Li S, Li G, Zhang S, Tang X, Ni S, Jian X, Xu C, Zhu J, Lu M. The role of extracellular vesicles in mediating progression, metastasis and potential treatment of hepatocellular carcinoma. Oncotarget. 2017 Jan 10;8(2):3683-3695. doi: 10.18632/oncotarget.12465. PMID: 27713136; PMCID: PMC5356911.
3. Abudoureyimu M, Zhou H, Zhi Y, Wang T, Feng B, Wang R, Chu X. Recent progress in the emerging role of exosome in hepatocellular carcinoma. Cell Prolif. 2019 Mar;52(2):e12541. doi: 10.1111/cpr.12541. Epub 2018 Nov 5. PMID: 30397975; PMCID: PMC6496614.
4. Zhao L, Shi J, Chang L, Wang Y, Liu S, Li Y, Zhang T, Zuo T, Fu B, Wang G, Ruan Y, Zhang Y, Xu P. Serum-Derived Exosomal Proteins as Potential Candidate Biomarkers for Hepatocellular Carcinoma. ACS Omega. 2021 Jan 4;6(1):827-835. doi: 10.1021/acsomega.0c05408. PMID: 33458533; PMCID: PMC7808137.
5. Li S, Chen L. Exosomes in Pathogenesis, Diagnosis, and Treatment of Hepatocellular Carcinoma. Front Oncol. 2022 Jan 27;12:793432. doi: 10.3389/fonc.2022.793432. PMID: 35155236; PMCID: PMC8828506.
6. Chen W, Mao Y, Liu C, Wu H, Chen S. Exosome in Hepatocellular Carcinoma: an update. J Cancer. 2021 Mar 5;12(9):2526-2536. doi: 10.7150/jca.54566. PMID: 33854614; PMCID: PMC8040701.
7. Sasaki R, Kanda T, Yokosuka O, Kato N, Matsuoka S, Moriyama M. Exosomes and Hepatocellular Carcinoma: From Bench to Bedside. Int J Mol Sci. 2019 Mar 20;20(6):1406. doi: 10.3390/ijms20061406. PMID: 30897788; PMCID: PMC6471845.
8. Muñoz-Hernández R, Rojas Á, Gato S, Gallego J, Gil-Gómez A, Castro MJ, Ampuero J, Romero-Gómez M. Extracellular Vesicles as Biomarkers in Liver Disease. Int J Mol Sci. 2022 Dec 19;23(24):16217. doi: 10.3390/ijms232416217. PMID: 36555854; PMCID: PMC9786586.
9. Wei XC, Liu LJ, Zhu F. Exosomes as potential diagnosis and treatment for liver cancer. World J Gastrointest Oncol. 2022 Jan 15;14(1):334-347. doi: 10.4251/wjgo.v14.i1.334. PMID: 35116120; PMCID: PMC8790408.
So, the question is what is new in the review manuscript “Exosomes as biomarkers for diagnosis and prognosis of hepatocellular carcinoma” by Yulai Zeng et al compared to the already published reviews?
Reviewer 3 Report
The review submitted by Kang He, Yi Luo, and the co-authors is devoted to the exosomes as biomarkers of hepatocellular carcinoma
The reviewer has some questions and has found some issues that should be resolved prior the manuscript being recommended for publication in Pharmaceutics
1) Typos
Fig. 1 - mi_RNA -> miRNA; cricRNA -> circRNA
see line 82 "Figure legend" - should be removed
2) Whether the Authors consider that DNA is one of the "Major components of exosomes" - see line 88 and line 70?
3) Since an undoubted mess in different Extracellular Vesicles (EVs) has been observed in the scientific literature, and MISEV2018 recommendations are perhaps the only relevant authority source for the classification of various EVs, these recommendations should be critically described and reviewed in any review devoted to the exosomes (or other EVs).
First of all, the authors should describe specific exosomal markers and methods used to prove EV-nature of vesicles and specific markers related to the exosomes. Second, a critical position to articles describing exosomes related to the HCC (which methods were used for isolation, evidence that the material is exosomes etc., in the case of all papers that are used as References, should be provided.
Since most of the papers use a sole method of isolation, unfortunately - commonly, a single ultracentrifugation is used; most of the results describe crude sediments, and the exosomal fraction is highly contaminated by co-precipitated proteins, nucleic acids, components of cell membranes etc.
4) The physiological significance of exosomes in
- biological regulation of HCC cells
- metastasis and progression of HCC
- immunosuppression of HCC
should be described.
The physiological role of exosomal components in the development of HCC
- miRNA
- lncRNA
- proteins
should be described as well.
In its current state, the manuscript contains various facts related to the exosomes and their components without a proper critical review of physiological significance, which is relevant to the reader and is relevant for the literature review.
5) Also, I recommend that the authors emphasize the possible physiological significance of exosomes and their particular components in the development of HCC in the "Discussion".
I hope the resolution of these comments will improve the article and make it more cited by potential readers.
Sincerely
Round 2
Reviewer 1 Report
No more comments
Reviewer 2 Report
The manuscript can be published
Reviewer 3 Report
The review, submitted after the R1, has been significantly improved according to the referees' comments. The manuscript might be accepted for publication in Pharmaceutics in its current state.
Sincerely,